# Augmented reality-based affective training for improving care communication skill and empathy

**Atsushi Nakazawa**[1]*, **Miyuki Iwamoto**[2], **Ryo Kurazume**[3], **Masato Nunoi**[4], **Masaki Kobayashi**[5], **Miwako Honda**[6]

1 Graduate School of Interdisciplinary Science and Engineering in Health Systems, Okayama University, Okayama, Japan, 2 Department of Advanced Fibro-Science, Kyoto Institute of Technology, Kyoto, Japan, 3 Faculty of Information Science and Electrical Engineering, Kyushu University, Fukuoka, Japan, 4 School of Human Sciences, Sugiyama Jogakuen University, Nagoya, Aichi, Japan, 5 Division of geriatric medicine, Rochester Regional Health System, Rochester, NY, United States of America, 6 Division of Geriatric Research, National Hospital Organization Tokyo Medical Center, Tokyo, Japan

* nakazawa.atsushi@okayama-u.ac.jp

**Data Availability Statement:** Data cannot be shared publicly because of the approved research proposal granted by the ethics committee of Tokyo Medical Center. Data are available from the

## Abstract

It is important for caregivers of people with dementia (PwD) to have good patient communication skills as it has been known to reduce the behavioral and psychological symptoms of dementia (BPSD) of PwD as well as caregiver burnout. However, acquiring such skills often requires one-on-one affective training, which can be costly. In this study, we propose affective training using augmented reality (AR) for supporting the acquisition of such skills. The system uses see-through AR glasses and a nursing training doll to train the user in both practical nursing skills and affective skills such as eye contact and patient communication. The experiment was conducted with 38 nursing students. The participants were assigned to either the Doll group, which only used a doll for training, or the AR group, which used both a doll and the AR system. The results showed that eye contact significantly increased and the face-to-face distance and angle decreased in the AR group, while the Doll group had no significant difference. In addition, the empathy score of the AR group significantly increased after the training. Upon analyzing the correlation between personality and changes of physical skills, we found a significant positive correlation between the improvement rate of eye contact and extraversion in the AR group. These results demonstrated that affective training using AR is effective for improving caregivers' physical skills and their empathy for their patients. We believe that this system will be beneficial not only for dementia caregivers but for anyone looking to improve their general communication skills.

## Introduction

Caregiving for people with dementia (PwD) is becoming increasingly important as the number of people with dementia has been growing worldwide [1, 2]. In Japan, the number of PwD is expected to exceed seven million in 2025, and the number of caregivers that will be needed is estimated to be 2.53 million. However, the number of actual caregivers has been decreasing

Institutional Data Access / Ethics Committee contact via email (215-rinri@mail.hosp.go.jp), or from author Miwako Honda M.D via email (honda-1@umin.ac.jp), for researchers who meet the criteria for access to confidential data.

**Funding:** This work was supported by JST CREST Grant Number JPMJCR17A5. The funders had no role in study design, data collection and analysis, decision to publish, or preparation of the manuscript.

**Competing interests:** The authors have declared that no competing interests exist.

and is estimated to be only 2.15 million in 2025 [3]. One of the causes of the decrease in caregivers is burnout. Dementia can cause symptoms similar to those of mental illness, known as behavioral and psychological symptoms of dementia (BPSD), which can make it challenging for caregivers to handle their patients. As a result, more and more caregiving staff members and family caregivers are experiencing burnout and leaving the profession [4].

Improving nursing communication skills is a potential solution to the aforementioned problems [5–9]. In [5], studies about communication training and its effects on carer and care-receiver outcomes in dementia settings were reviewed. The result show the communication training affects strong positive impact on both carer outcomes and care-receiver outcomes. Specially, the intervention effects were found to be strongest on carer communication skills, knowledge and attitudes, and sustained beyond the intervention period for several months. They also pointed out that the design of these interventions is an important factor, namely, the interventions consist of demonstration and/or presentation of the targeted knowledge and skills through lectures, followed by discussion and practice in an interactive face-to-face environment. In [10], 116 medical students participated in a communication skills training program, and their empathy scores were followed up for six years. Their scores significantly increased after the training but did not remain at the same level six years later. In [9], a four-week care communication skills training program was conducted, which included video lectures and bedside instruction. The mean empathy scores and the personal accomplishment burnout scores of the participants significantly increased after training.

We address the lack of communication training and follow-up by using augmented reality (AR) technology. In recent years, opportunities for person-to-person or on-site training have been decreasing, especially due to COVID-19. Therefore, skill training using VR and AR technology is becoming much common. It is also efficient and cost-effective for trainees to brush up their skills by self-training with the AR-based systems. Applications of such systems include assembly tasks [11], primary education [12], and physical education in school [13]. However, the most common area of application is health and medicine [14], namely for training physicians and nurses in manipulation skills [15–17]. While most studies are pre-post experiments with one group, the study in [17] compared VR with traditional lecture-based methods for triage lessons and found the user's subjective experience was improved while no improvement was found in paper-based test scores. Another important area is social skills training for developmental disorders such as autism spectrum disorder (ASD). VR and AR systems contribute to learning reinforcement for students with specific educational support needs, improving the possibilities for establishing social ties. According to recent intervention studies [18], VR/AR-based intervention improved the social skills of participants with ASD; however, the methodological limitations such as the lack of control groups, follow-up measures, and generalization of the results should be acknowledged.

In this study, we developed an AR-based nursing communication training system and evaluated its effectiveness. As a reference for ideal care communication, we use the caregiving methodology *Humanitude* developed by Geneste and Marescotti [19], which has been introduced in more than 600 hospitals and nursing homes in Europe. Several studies have reported the effectiveness of Humanitude in reducing BPSD and the burden on caregivers [20]. In addition to verbal communication, the Humanitude system also focuses on non-verbal communication techniques such as eye contact, face-to-face distance, and physical touch [21], which are often difficult to learn from classroom lectures only.

Our research aim is to evaluate the effectiveness of AR for affective communication training for caregivers. To our knowledge, our study is the first to evaluate AR-based affective communication training using randomized controlled trials. We hypothesize that our skill-training AR system will respond appropriately to the participants, enabling them to improve their

physical skills (e.g., face-to-face distance, eye contact, conversation) more effectively than other methods as well as psychological state (e.g. empathy to the patients).

To detect physical measures, face-to-face communication was recorded with a head-mounted first-person camera, and the features were automatically detected by using AI-based image analysis. Regarding the change of the psychological state, we assessed the level of empathy towards the patient before and after the training. As previous studies have shown that learning using Humanitude over time improves empathy, we investigated whether similar results can be observed with AR and whether it is more effective than regular training.

The second research aim is to investigate the relationship between the training results and participants' personalities. As social affective-communication is known to be related to personality, we aim to investigate how the improvement in communicative behavior through training relates to one's personality.

The remainder of the paper is as follows. In section 2, the study design, study procedure, the AR system and the data analysis are described, followed by the results and discussion.

## Study design and participants

A two-group randomized controlled study was conducted with participants recruited from students in the clinical nursing department at Bukkyo University, Japan. The demographics of the participants (N = 38) are listed in Table 1. Participants ranged from 18 to 28 years old, with an average age of 20.9 years old. All of them were born in Japan and are native Japanese speakers. The students ranged from first to fourth years, and 82% of them were female. All students had average computing literacy for their age group, but no experience using VR/AR systems and Humanitude care methodology. They were paid 2,000 JPY (about 16 USD) for their two-hour participation. Written informed consent was obtained from all patients for the participation in the study and publication of the accompanying images. The individual pictured in Fig 1 has provided written informed consent (as outlined in PLOS consent form) to publish their image alongside the manuscript. They were randomly assigned to either the control (Doll) or the experimental (AR) group.

## Measures

All participants filled out and submitted a shortened version of the Japanese Big-Five Scale [22] and the Jefferson Scale of Empathy (JSE) Health Profession Students' version [23]. A 7-point Likert scale was used for both surveys. The Big-Five was administered and collected once before the experiment, whereas the JSE was collected before and after the experiment to observe the effects of training on the participants' empathy.

To measure the physical behavioral features, first-person videos were recorded from head-mounted cameras worn by the participants. From the videos, we then retrieved face-to-face

**Table 1. Participants.**

|  | Control (Doll) | | Experimental (AR) | |
|---|---|---|---|---|
|  | M | F | M | F |
| All | 3 | 14 | 4 | 17 |
| 1st grade | 0 | 2 | 0 | 5 |
| 2nd grade | 0 | 3 | 0 | 3 |
| 3rd grade | 0 | 3 | 2 | 4 |
| 4th grade | 3 | 6 | 2 | 5 |
| Av. age | 21.06 | | 20.81 | |

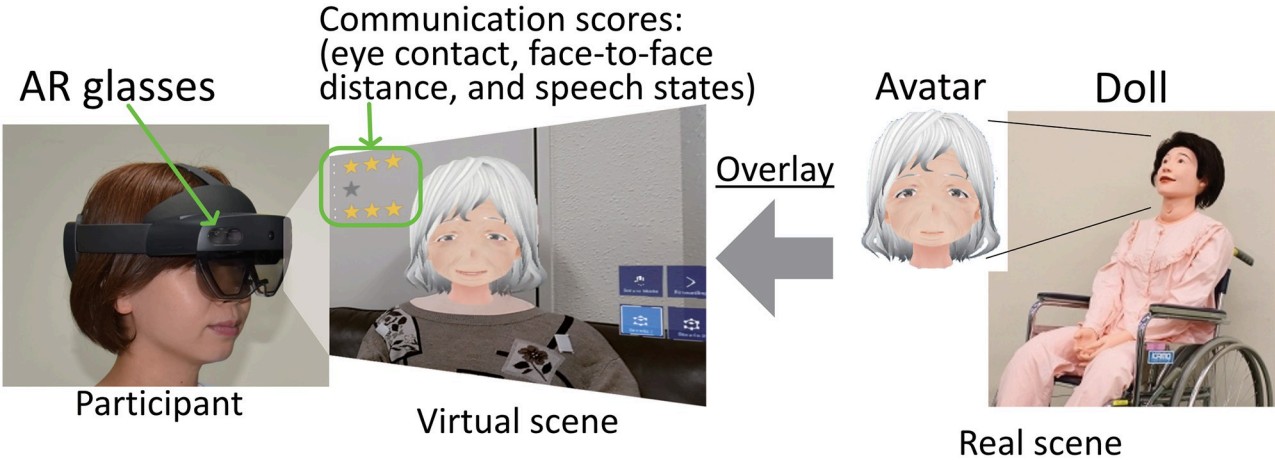

**Fig 1. System configuration of HEARTS.**

distance and pose, the occurrence of eye contact between the participants and simulated patients, and the length of the caregiver's speech. These skills are vital aspects of the Humanitude care technique.

## Apparatus

**Simulated patients.** Twelve female faculty members in the nursing department at Bukkyo University participated as simulated patients. Before the experiment, they attended a three-hour lecture on portraying a simulated patient with dementia. The participants' training groups were not disclosed to the simulated patients.

**Life-size nursing doll.** Life-size nursing dolls (Sakura, Kyoto Kagaku) were used for training. The doll was positioned sitting in a wheelchair, and the participants practiced communication and assigned tasks in independent practice sessions. For the AR group, the face of the doll was covered by a white cloth because the face was presented by a AR headset, whereas it was not covered for the Doll group.

**First- and third- person video.** The Tobii Pro Glasses 3 were used to collect the first-person video and gaze data of the participants while they were simulating patient care tasks in the pre- and post-evaluations. In addition, a standard video camera was used to record the sessions. Although the Tobii systems collected the participants' gaze data, we did not use the data because the number of frames where the data were successfully taken varied between individuals due to technical limitations. Thus, instead of using the gaze data, we used the first-person videos (frontal view) for further analysis.

**AR system.** The AR-based care communication training system HEARTS (Humanitude AR Training System) developed by Kurazume et al [24] was used to train the participants in the AR group. The system configuration of HEARTS is shown in Fig 1. The system consists of an AR device (Microsoft HoloLens 2) and the nursing doll. To imitate the interaction between a patient and a caregiver, we superimposed a three-dimensional CG model of a patient's face onto the head of the doll using AR technology. The gaze direction, blinking, and facial expressions of the simulated patient were dynamically controlled on the basis of the caregiver's movement. The display content and avatar's motion were created in a Unity 2020 environment. The flowchart of the avatar's response is shown in Fig 2. Three facial expressions corresponding to happy, angry, and neutral were implemented as shown in Fig 3. The eye contact

**Fig 2. Flowchart of the avatar's motion in HEARTS, where $T_i$ = 15 deg, $T_d$ = 700 mm, and $T_a$ = 25 deg.**

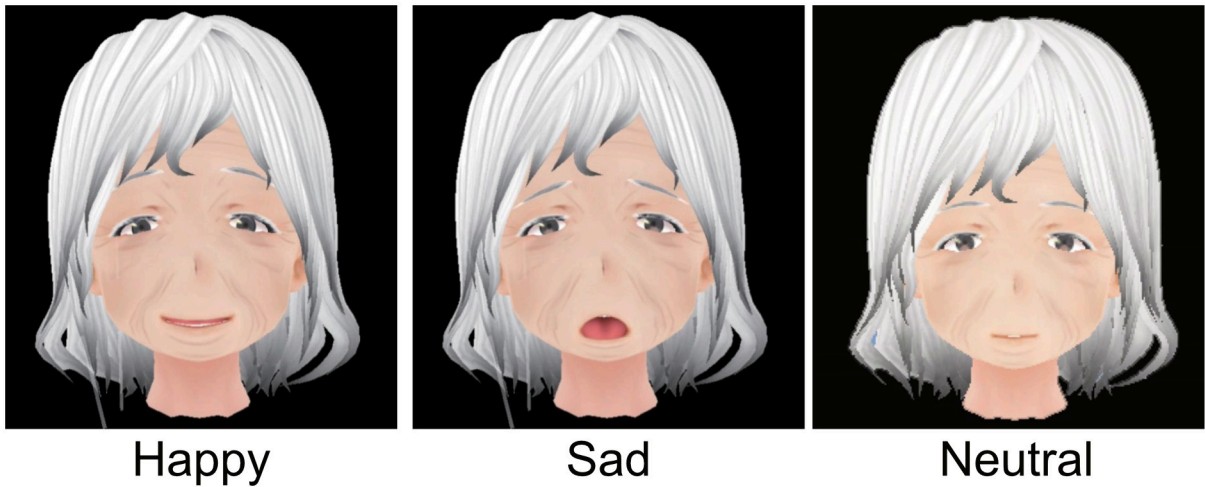

**Fig 3. Facial expressions of avatar.**

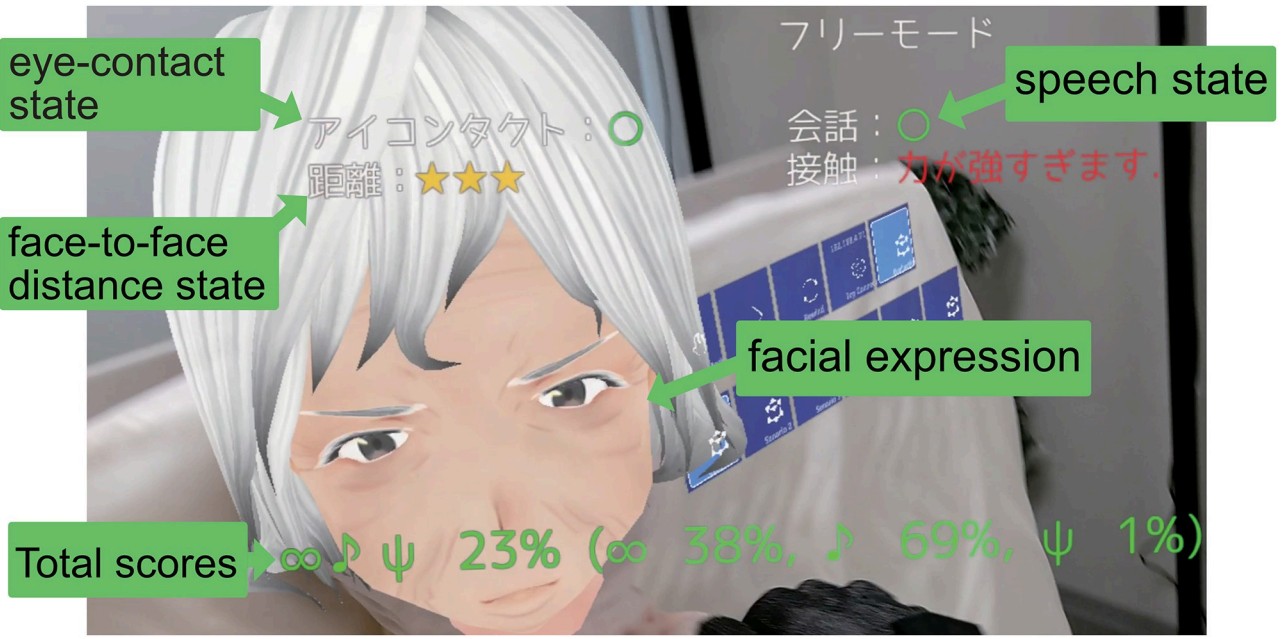

**Fig 4. Gaze, speech scores displayed on HEARTS.**

and verbal communication between the patient and the caregiver were detected and evaluated online by the AR device. The evaluation results are indicated by "Good," "Bad," or the number of asterisks on the screen of the HoloLens. The criteria is as follows:

- **Eye contact**
  "Good" if the frontal axis of the HoloLens passes through the AR's face and the relative angle between the frontal axis of the HoloLens and the avatar's frontal axis is less than 25 degrees, otherwise "Bad." The facial expression also changes to "happiness" if the eye contact is "Good."

- **Verbal communication**
  "Good" if the sound pressure is stronger than the threshold, otherwise "Bad.

- **Face-to-face distance**
  One, two, or three asterisks corresponding to distances over 1400 mm, between 700 and 1400 mm, and under 700 mm, respectively.

If there is no eye contact or conversation, a "No communication" warning is issued and the facial expression changes to "sad" or "fear." In addition, the numerical score for eye contact is evaluated by the ratio of the total time the caregiver makes eye contact to the total nursing care time. The numerical score of the verbal communication skill is determined by the ratio of the total speaking time to the total nursing care time. These scores are displayed on the screen of the HoloLens2 as shown in Fig 4.

## Procedure

The evaluation was conducted in accordance with the procedure approved by the ethics committee of the Tokyo Medical Center numbers R21-055 and R21-079. The study protocol

adheres to the Declaration of Helsinki and follows the latest ethical guidelines for clinical research in Japan, and is registered in the Japanese clinical research registry system that meets the criteria of the International Committee of Medical Journal Editors (ICMJE) (University Hospital Medical Information Network Clinical Trials Registry Number: UMIN000046670).

The overview of the procedure is shown in Fig 5. Two days before the experiment, the participants were asked to read a Humanitude textbook, which provides a brief overview of the fundamental aspects and skills of Humanitude, and then they answered the JSE and Big-Five through online questionnaires. Afterwards, the participants answered a ten-question mini test about Humanitude to encourage them to learn about Humanitude from the textbook.

On the day of the experiment, the participants were first asked to perform care tasks for a simulated patient with dementia, e.g., the patient did not give clear responses to the participants. The care tasks included helping the patient change clothes and put on socks. During this experiment, the participants wore an eye tracking device, and the simulated patient wore a first-person camera. The entire trial was recorded by a video camera. Then the participants were randomly assigned to either the control or experimental group. The participants were asked to practice the same task by themselves for 30 minutes. In the control group, the participants practiced the task using a patient-care doll, while the participants assigned to the experimental group used the AR headset and the doll with the facial image overlaid on its head. After the independent training session, the participants performed the same tasks with the same simulated patient. Finally, they filled out the JSE and a questionnaire on the feasibility and usability of the training.

Before experiment
Read Humanitude textbook,
answer questionnaires (Personality and Empathy)

Experiment day
Pre-evaluation: Perform care-tasks
for a simulated patient

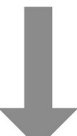
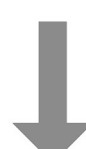

**Control group** : Training using a doll     **Experimental group** : Training using AR

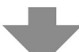
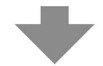

Post-evaluation : Perform care tasks
for the simulated patient.

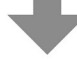

Questionnaires (Empathy)

Fig 5. Overview of experiment.

### Analysis of first-person video

We used the first-person videos from the cameras worn by participants to retrieve eye contact and face-to-face posture information, which is important for evaluating the Humanitude skills [25]. Although we obtained the participants' gaze data, we did not use the information because the success rates of the data collection varied greatly between individuals. We used the video starting from the beginning (knocking the door) to the end of the care sessions (leaving from the room) for the analysis.

**Face-to-face distance and pose.** The face-to-face distance and pose (angle) between a participant and a simulated patient were obtained using the image recognition (facial parts detection) algorithm illustrated in Fig 6. First, the facial parts of a simulated patient were detected using Amazon Rekognition, and then the distance and angle to the face were obtained using Perspective-n-Point (PnP) pose computation [26]. Here, camera parameters of the first-person cameras and predefined facial size were used. From the analysis, we obtained (1) the facial detection rate that represents how likely the participant looked at the simulated patient's face, (2) face-to-face distance, and (3) roll, pitch, and yaw angles.

**Eye contact.** The eye contact was measured by using a deep neural network-based eye contact detection algorithm [27, 28], which has been shown to perform as accurately as human experts. We inputted the first-person video and facial regions to the algorithm and obtained eye contact scores. We assumed the frame where the output eye contact score was greater than 0.8 as the engagement of eye contact.

**Participant speech.** The length of the participants' utterances were detected from the video data. We manually annotated the video and obtained the total utterance duration of each session.

## Results

We compared the JSE scores, face-to-face distance and pose, the rate of eye contact, and length of caregivers' speech between the pre- and post-evaluations. The paired t-test (single sided) was used for statistical evaluations, and the error bars in figures denote standard deviation.

### Empathy

Fig 7 shows the JSE scores before and after the training sessions. Note that the scores ranged from 20 (lowest) to 140 (highest). The average score of all participants significantly increased

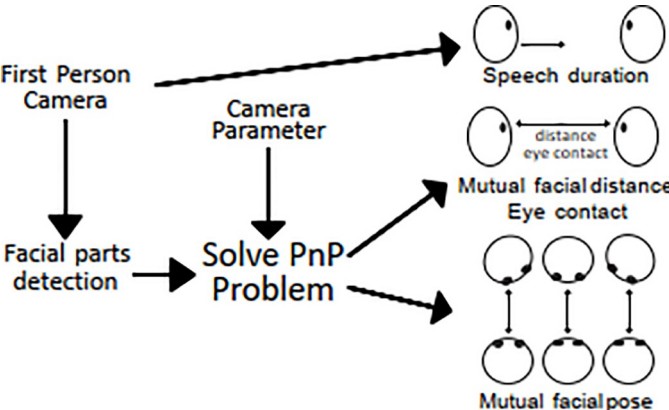

**Fig 6. Estimation of face-to-face distance and pose (angles).**

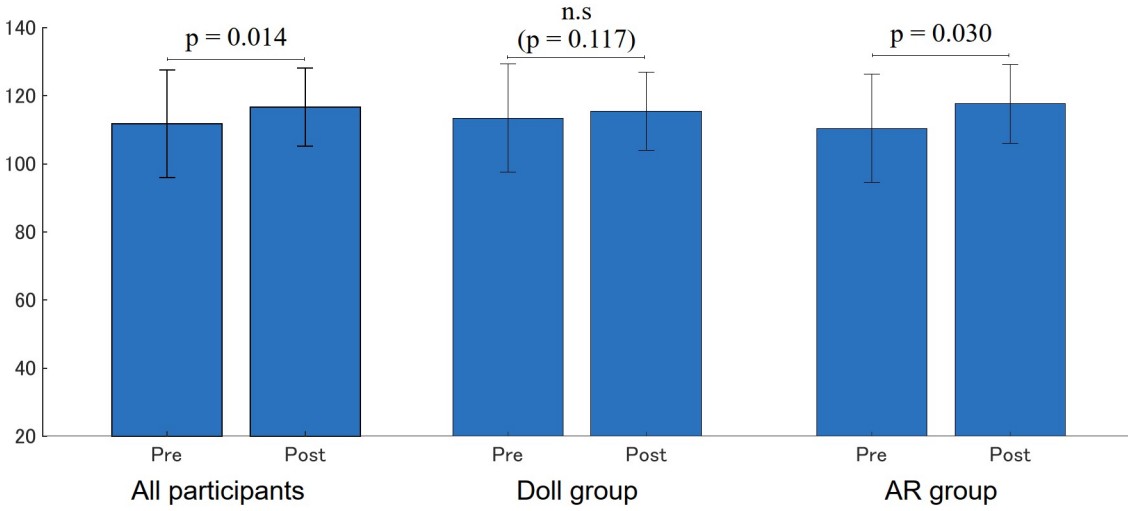

**Fig 7. Jefferson Scale of Empathy scores before and after training sessions.**

(Pre: 111.79 (15.87), Post: 116.68 (11.53), p = 0.014) and that of the Doll group had no significant change (Pre: 113.47 (13.13), Post: 115.47 (12.84), p = 0.117), whereas that of the AR group significantly increased (Pre: 110.43 (17.99), Post: 117.65 (10.58), p = 0.030)).

## Face-to-face distance and pose

Fig 8 shows the face detection rates which indicate how frequently the participant looked at the simulated patient. The face detection rate significantly increased (Pre: 0.157 (0.105), Post: 0.184 (0.110), p = 0.012) in all participants, and that of the AR group also significantly increased (Pre: 0.168 (0.122), Post: 0.212 (0.122), p = 0.011). However, no significance was found in the Doll group (Pre: 0.145 (0.080), Post: 0.150 (0.085), p = 0.323).

As shown in Fig 9, the face-to-face distance significantly decreased in all participants (Pre: 980.77 (445.28)[mm], Post: 899.35 (369.48)[mm], p = 0.045), but no significance was found in

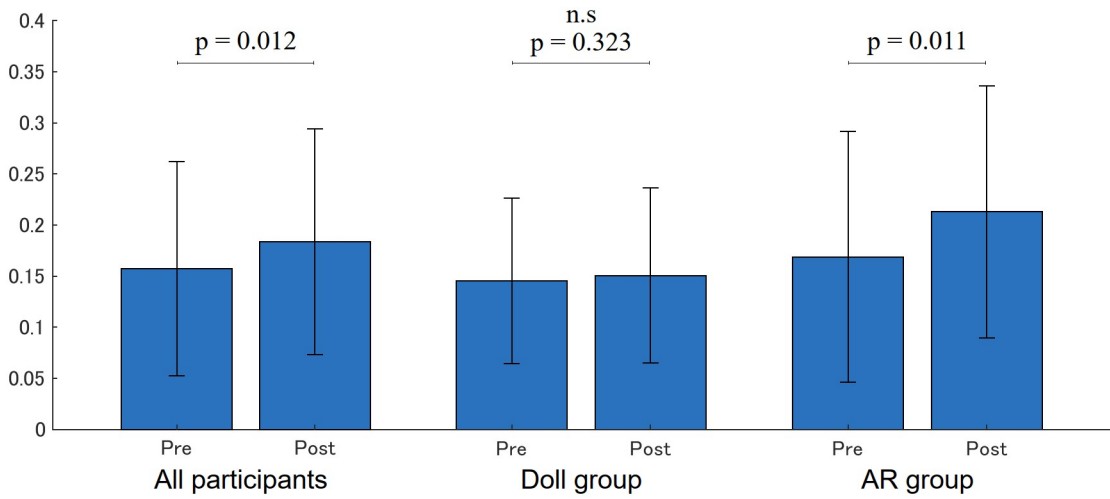

**Fig 8. Face detection rate.**

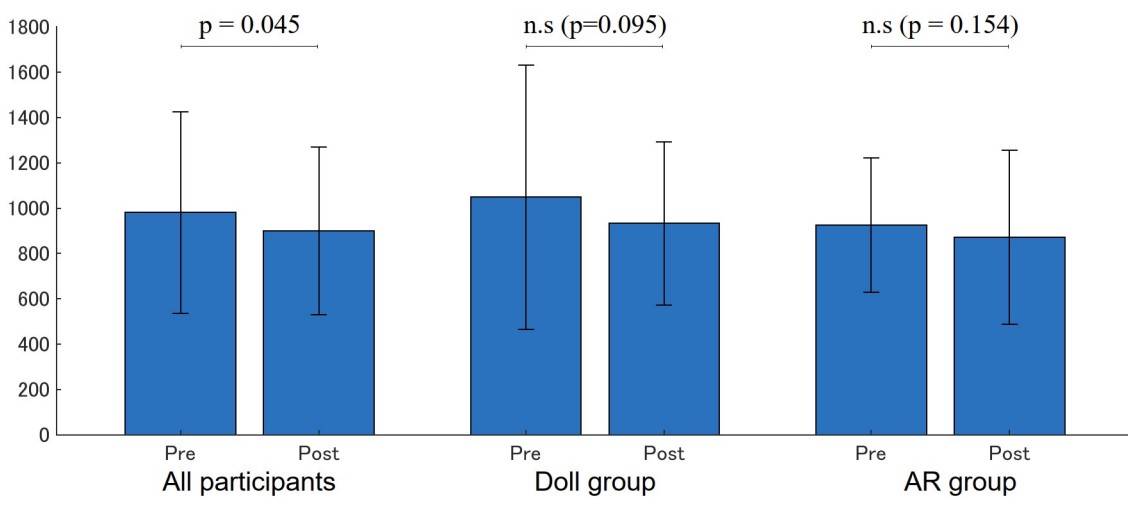

**Fig 9. Face-to-face distance ([mm]).**

the Doll group (Pre: 1048.96 (582.35)[mm], Post: 933.32(359.49) [mm], p = 0.095) or the AR group (Pre: 925.58 (297.09)[mm], Post: 871.84(383.92)[mm], p = 0.153).

Fig 10 shows the occurrence rates of when the face-to-face distance was less than 700 [mm], which is the same threshold used in the AR system. They significantly increased in all participants (Pre: 0.096 (0.080), Post: 0.122 (0.087), p = 0.001) and in the AR group (Pre: 0.109 (0.093), Post: 0.145 (0.096), p = 0.002), but no significance was found in the Doll group (Pre: 0.081 (0.058), Post: 0.094 (0.068), p = 0.108).

Fig 11 shows the results of face-to-face posture (angles). The mean yaw-angles significantly decreased in all participants (Pre: 17.19 (8.41)[deg], Post: 15.58(6.39) [deg] p = 0.040) and in the AR group (Pre: 18.16 (9.35)[deg], Post: 15.63 (6.73) [deg], p = 0.010), while no significance was found in the Doll group (Pre: 15.98(7.18)[deg] to 15.51(6.14)[deg], p = 0.382). The smaller yaw angle indicated that participants looked at the simulated patients' face in the frontal direction, leading to improved face-to-face communication skills for the participants in the AR group.

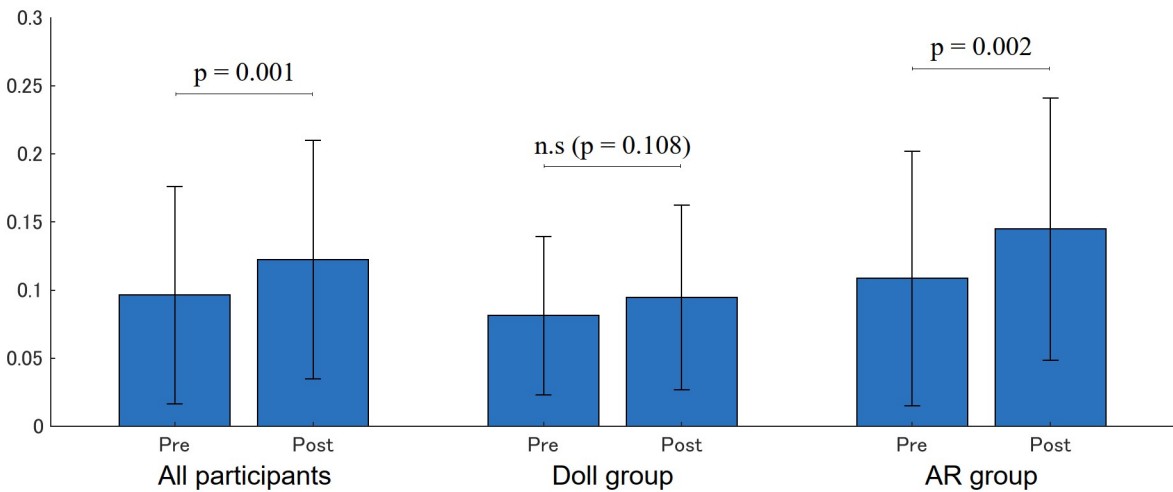

**Fig 10. Occurrence rate of where face-to-face distance is less than 700 [mm].**

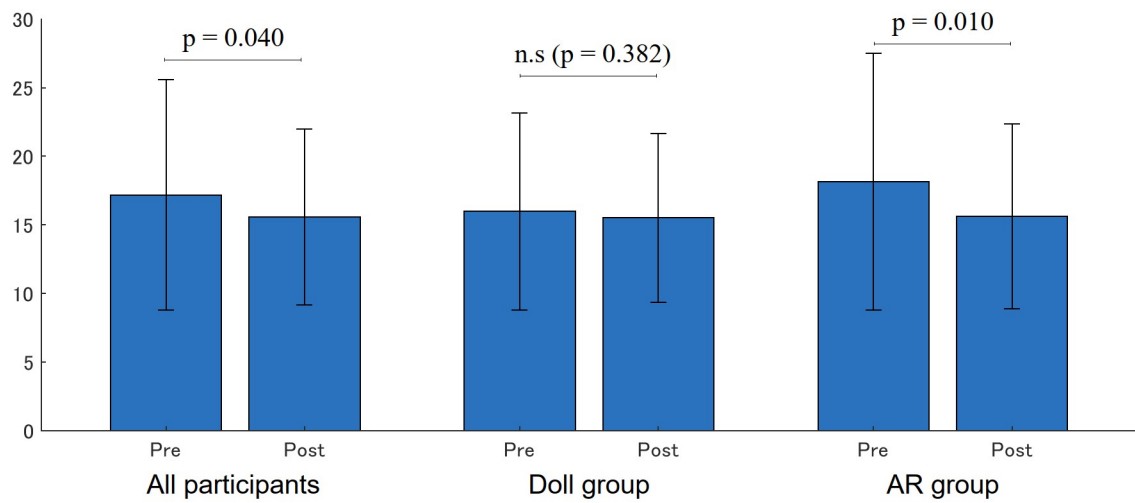

**Fig 11. Face-to-face angle (yaw rotation [deg]).**

### Eye contact

Fig 12 shows the occurrence rate of eye contact which is computed as the duration of eye contact divided by the session length. A significant increase was found in the AR group (Pre: 0.023 (0.025), Post: 0.032 (0.029), p = 0.039), while no significance was found in the Doll group (Pre: 0.016 (0.016), Post: 0.015 (0.018), p = 0.501).

### Speech

Fig 13 shows the utterance duration normalized by the session length. A significant increase was found in both the Doll and AR groups (Doll—Pre: 0.141 (0.073), Post: 0.221 (0.098), p = 0.0005; AR—Pre: 0.147 (0.066), Post: 0.267 (0.102), p = 0.000).

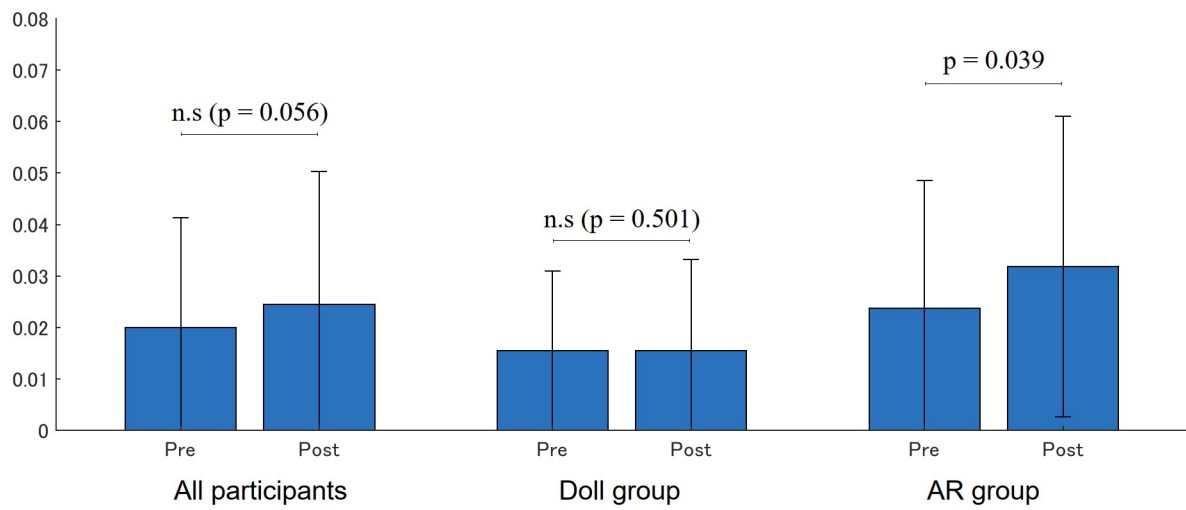

**Fig 12. Occurrence rate of eye contact.**

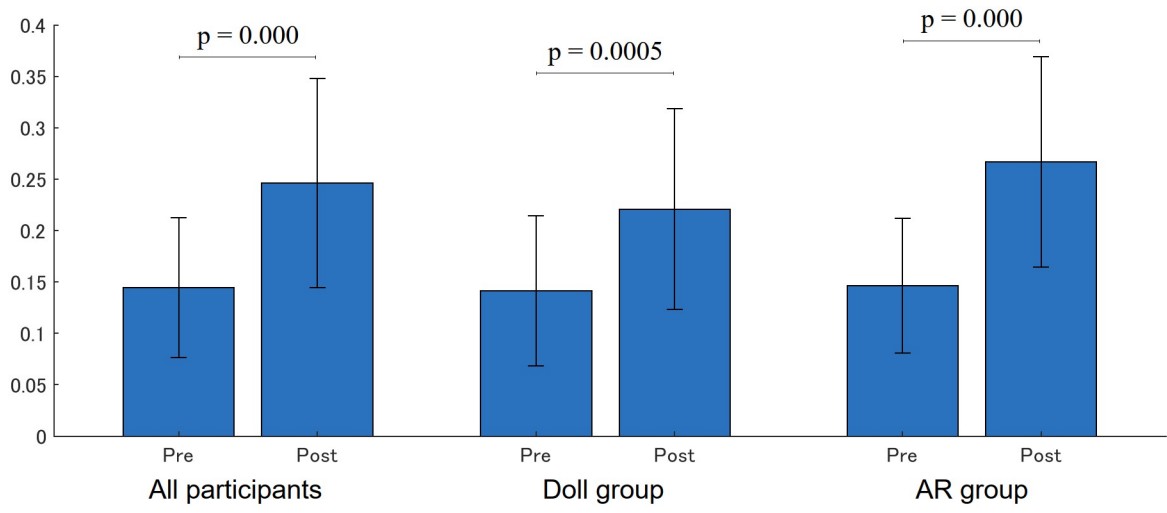

**Fig 13. Utterance.**

## Personality and behavior

Next, the correlation between personality and physical behavioral parameters was examined. A significant correlation was found between extraversion (ranging from 5 to 35) and the occurrence of eye contact (ranging from 0.0 to 1.0) as shown in Fig 14. Specifically, a significant correlation was found in the post-evaluation in all participants (Pre: $r = 0.198$ (p = 0.230), Post: $r = 0.333$ (p = 0.041)) and in the AR group (Pre: $r = 0.278$ (p = 0.210), Post: $r = 0.482$ (p = 0.023)), but no significance was found in the Doll group (Pre: 0.069 (p = 0.792), Post: $r = 0.117$ (p = 0.654)).

## Feasibility and acceptability

In response to the prompt "The training time was adequate," 95% of the participants in the AR group answered "strongly agree" or "agree," in contrast to 59% of the participants in the Doll group. In response to "I felt as if I was seeing the actual patient in front of me," 90% of the

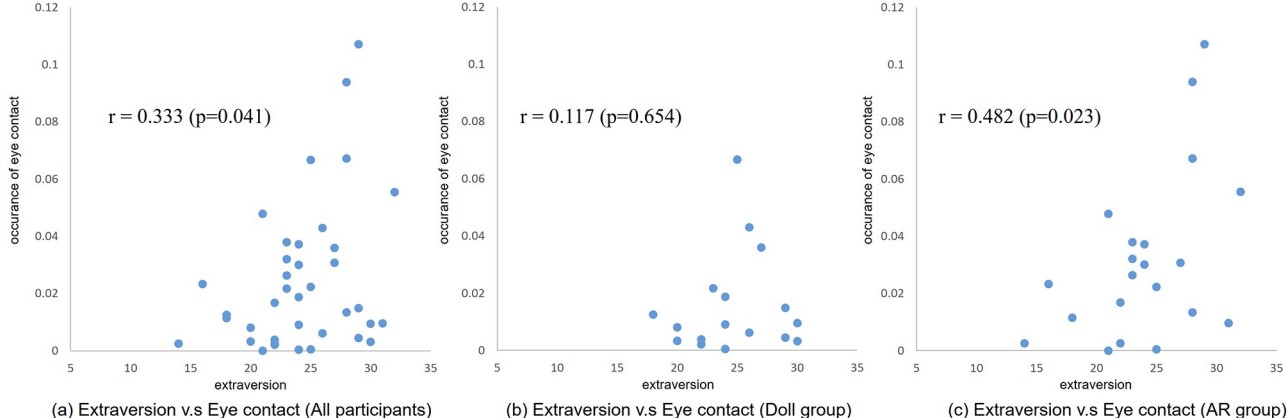

**Fig 14. Correlation between the extraversion and occurrence rate of eye contact in the post-evaluation.** (a) Extraversion v.s Eye contact (All participants). (b) Extraversion v.s Eye contact (Doll group). (c) Extraversion v.s Eye contact (AR group).

participants in the AR group and 70% of the participants in the Doll group answered "strongly agree" or "agree."

In the open-ended response after the training, participants in the AR group expressed that they became more aware of the importance of continuous communication, e.g. making eye contact and speaking with the patient. They stated, "I thought it was good that the system let me know when I was not communicating well during care," "The system told me when my eye contact was even a little off, so I realized that I was not actually making eye contact even though I thought I was," "I became more conscious of making eye contact with patients and communicating more during care," and "I realized that communication during care is important." The responses to the general impact of the AR system were largely positive. For example, one participant wrote, "It was good that the system informed me when I was not communicating with the patient." They also shared what they would like to see improved for individual practice; for example, "The patient's only utterance was laughter, so it would have been nice to have a few more statements or changes in emotion," "It was good to be able to check the evaluation in real-time but I wish there had been some model answers or hints on how to get a higher evaluation in learning."

## Discussion

In this study, the impact of the AR-based affective training for nursing students was evaluated by means of a two-group randomized controlled study. The overall results indicate that the system had a more of a positive impact on the participants in the AR group than those trained by the existing (Doll) method. In particular, the frequency of the patient's faces appearing in the first-person video, the occurrence of smaller face-to-face distances, and the occurrence of eye contact with the patient significantly increased after the training, while they did not increase significantly in the Doll group. These parameters indicated that the participants adjusted themselves to look at the patients' faces and eyes more frequently at a closer distance, which is one of the important skills in Humanitude skill system and dementia care. At the same time, the face-to-face yaw angle was also significantly decreased in the AR group, which means the participants tended to looked at the patient's face in the frontal direction. This is another important skill factor in dementia care as PwDs become difficult to identify from non-frontal faces [29]. These findings demonstrate that the AR-based affective training is effective for learning physical skills needed in dementia care, such as making eye contact, decreasing face-to-face distance, and facing the direction of the patient.

Interestingly, although the system was designed so that participants were in the role of a caregiver rather that PwD, the empathy score significantly increased in the AR group while it did not in the Doll group. This may be because the AR group received affective feedback from the avatar through positive and negative facial expressions and laughter, whereas the Doll group did not receive any feedback. The findings indicate that compared with the conventional training method, the AR-based skill training system had a greater impact on the participants' acquisition of both physical and affective skills for dementia care.

The existing VR systems which enabled participants to experience life as PwD increased their empathy through a first-person experience. Our proposed system also increased the participants empathy even though it was a third-person experience because they could empathize with the avatars' responses, e.g., smiling, anger. We believe that the participants' empathy for PwD can be further increased by using both types of systems at the same time.

Another interesting finding is the correlation between extraversion and eye contact. In the first meeting with the simulated patient (Pre-evaluation), no significance was found, whereas a significant correlation was found in the AR group in the second meeting (Post-evaluation).

This indicates a positive relationship exists between extraversion and eye contact. Furthermore, the affective communication training increased eye contact, and participants with higher extraversion scores could effectively improve their eye contact skill by using the AR system.

The three major contributions of this paper can be summarized as follows. The first is the development of an AR-based system for affective training in health-care communication. The educational effectiveness of the system was clarified through a randomized controlled trial between the existing method and AR-based training groups. Although several AR-based training systems have been proposed in the past, they were designed for learning physical skills such as surgical techniques and therefore did not target communication skills with patients. As mentioned in the Introduction, training for affective communication with PwD should be effective in reducing BPSD. In [9], physicians were given an educational intervention on communication techniques, where instructors taught the communication skills. The basic communication skills of the physicians, such as eye contact, verbal expression, and touch, improved as a result. The proposed AR-based educational intervention has considerable potential to enable easier and less costly communication education and our results show that AR-based self-training is just as or more effective than self-training without AR. [10] pointed out that even if communication training is given to medical students, the effect (empathy) returns to its original state a few years after the training. In that sense, the AR system can serve as an efficient and cost-effective way for users to continue self-training and maintain their improved empathy.

The second contribution is that objective quantitative indicators, such as the distance between faces based on image recognition AI, were used to assess the educational effectiveness of affective training in order to evaluate communication skills. Existing studies in health communication have quantitatively analyzed communication skills on the basis of human video coding [30, 31]. Similarly, in existing intervention studies for children with ASD who practiced communication skills using VR/AR, the effectiveness was assessed by a questionnaire assessment such as the School Function Assessment (SFA) [32], motivational interviewing assessment scale (MIAS), and self-efficacy scale [33]). In contrast, [34] conducted an intervention with children with developmental disabilities while they exercised in a projection-mapped environment. They measured the effects of the intervention using the relative positioning of the children through video analysis of 360 degree camera images; however, they used only positional information in a large environment. To the best of our knowledge, our work is the first study to automatically evaluate changes in communication using quantitative behavioral factors such as eye contact and distance between faces between caregivers and receivers. Similar efforts have been made in our previous studies, such as a comparative study of communication skill between experts and novices [25] and a before-and-after comparative study of educational intervention [9], and the effectiveness of the indicators was determined to distinguish these groups. We believe that the fact that similar effects were observed in the AR intervention in this study supports both the validity of the AR education and the validity of the assessment indicators.

Our third contribution is that we clarify the relationship between AR-based communication training and changes in the subjects' internal state, specifically an improvement in empathy towards the patients. While AR and VR systems have been developed where participants can experience being a PwD [35–37], our AR system is the first one where the user experiences being the caregiver of a PwD. Our results demonstrated that the caregivers' empathy toward the patients significantly improved after using this system. While conventional affective training intervention improved empathy scores [38, 39], in our study, the training group using the conventional method (Doll-based training group) showed no significant improvement in empathy, whereas the training group using AR showed significant improvement. Although

careful consideration is required, this suggests that the developed AR system improved empathy more effectively than the conventional method.

Interestingly, a significant correlation was found between extraversion and amount of eye contact in the AR training group. A similar relationship has been reported in the literature [40, 41], though in our study, the significant increase was observed only in the AR training group. Social signaling skills such as eye contact may be improved in the higher extraversion group with appropriate intervention such as AR-based training.

## Limitations and future work

Our study was conducted with 38 participants, which may seem small, but existing VR/AR studies have had similar or smaller numbers [18], making this a relatively large study of its kind. The study size is limited because of the cost of the AR system and the effort needed to operate it. For the same reason, the intervention in this study lasted only one day and was not a long-term intervention as in previous studies. We believe that the preliminary results demonstrate the need for larger scale and longer term intervention studies on AR technology. Caregivers' facial expressions are important when they communicate with PwDs; however, our system does not evaluate the participants' facial expressions. Similarly, the system does not evaluate the caregiver's physical touch with the patient or the content of the caregiver's speech. The caregiver's facial expressions, touch, and content of their speech will need to be integrated in the future to evaluate the quality of their caregiving skills for people with PwD. In this study, the participants were nursing students, but it is not clear whether the same results can be obtained with more experienced caregivers.

The avatar used in the study had a cartoon-like face so that its facial expressions would be obvious to novices (nursing students). However, the relationship between the facial rendering style (cartoon-like or realistic) and the outcome may also need to be investigated in the future.

## Author Contributions

**Conceptualization:** Atsushi Nakazawa, Miwako Honda.

**Data curation:** Miyuki Iwamoto.

**Formal analysis:** Atsushi Nakazawa, Miyuki Iwamoto.

**Funding acquisition:** Atsushi Nakazawa.

**Methodology:** Atsushi Nakazawa, Ryo Kurazume, Masato Nunoi, Masaki Kobayashi, Miwako Honda.

**Project administration:** Atsushi Nakazawa.

**Software:** Atsushi Nakazawa, Ryo Kurazume.

**Supervision:** Atsushi Nakazawa.

**Visualization:** Atsushi Nakazawa.

**Writing – original draft:** Atsushi Nakazawa.

**Writing – review & editing:** Atsushi Nakazawa.

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
