## [Decision Letter · Decision Letter 0]

1 Feb 2023

PONE-D-22-35280The Augmented Reality affective training increases the care communication skill and empathyPLOS ONE

Dear Dr. Nakazawa,

Thank you for submitting your manuscript to PLOS ONE. After careful consideration, we feel that it has merit but does not fully meet PLOS ONE’s publication criteria as it currently stands. Therefore, we invite you to submit a revised version of the manuscript that addresses the points raised during the review process.

We look forward to receiving your revised manuscript.

Kind regards,

Humaira Nisar

Academic Editor

PLOS ONE

Journal Requirements:

"This work was supported by JST CREST Grant Number JPMJCR17A5."

"This work was supported by JST CREST Grant Number JPMJCR17A5."

4. Please remove your figures from within your manuscript file, leaving only the individual TIFF/EPS image files, uploaded separately. These will be automatically included in the reviewers’ PDF.

5. We note that Figures 1 to 4 in your submission contain copyrighted images. All PLOS content is published under the Creative Commons Attribution License (CC BY 4.0), which means that the manuscript, images, and Supporting Information files will be freely available online, and any third party is permitted to access, download, copy, distribute, and use these materials in any way, even commercially, with proper attribution. For more information, see our copyright guidelines: http://journals.plos.org/plosone/s/licenses-and-copyright.

a. You may seek permission from the original copyright holder of Figures 1 to 4 to publish the content specifically under the CC BY 4.0 license. 

Additional Editor Comments:

PLease refer to the reviewers comments and revise the manuscript accordingly.

Reviewers' comments:

Reviewer's Responses to Questions

**Comments to the Author**

1. Is the manuscript technically sound, and do the data support the conclusions?

Reviewer #1: Partly

Reviewer #2: Yes

Reviewer #3: Partly

2. Has the statistical analysis been performed appropriately and rigorously? 

Reviewer #1: Yes

Reviewer #2: Yes

Reviewer #3: No

3. Have the authors made all data underlying the findings in their manuscript fully available?

Reviewer #1: Yes

Reviewer #2: Yes

Reviewer #3: Yes

4. Is the manuscript presented in an intelligible fashion and written in standard English?

Reviewer #1: Yes

Reviewer #2: Yes

Reviewer #3: Yes

5. Review Comments to the Author

Reviewer #1: This study cannot be accepted at this stage. Therefore, I would like to propose the following extensive modifications, which can enhance the reliability and validity of the authors’ study:

1. Research questions, that drive the paper, should be built in the introduction from an ongoing and pertinent bibliography (up to 2022-23) and these should be of global interest and not focused on a particular local problem. Identifying a research gap is the most important by indicating in-text some newer references that are significant to your particular field of research.

2. The problem statement needs to be clear into Introduction. Does the relevant literature refer the potentials of using “conventional” instead of typical AR-supported activities to drive us there?

3. Some indicative literature review articles regarding the use of AR-supported learning that the same authors read maybe of great importance to readers in order to identify the research “gap” more easily without mention too many previous studies.

4. The authors need to provide any potentials regarding the use of AR settings. Are there any literature reviews to give us a point of view about this investigation? In other words, the authors need to “combine” any key word that investigate with any relevant study that can be integrated in-text as “background”.

5. Any information respecting participants’ background can be beneficial for readers to understand better the level of technological literacy can overcome this “novelty effect” using AR.

6. The authors should make explicit suggestions about how their study affects the design or use of AR systems. Is there something new about a particular theory, or is there evidence of theory advancement?

7. The added value of authors’ work is not clear in terms of proper and current research (up to 2022-23), because several references are not up-to-date.

8. Authors should answer the main research question in the conclusions and discussion. Please provide a reasonable need to read your work’s results than previous ones or simply answer what we learned compared with current, significant research (up to 2022 should be your work’s “significance”).

9. How general are your results and how do you believe that such findings have to be of global interest? Please relate these with your limitations and Discussion that is not exist. Why?

10. Are there any points of view related to the consequences of this study’s limitations that may have an impact on their findings?

11. Conclusions and limits are too short for such a study.

12. Practical and educational implications are not provided.

Reviewer #2: This is a well written paper that presents a relevant description of a new methodology. The authors useful information has been provided. this papare may be of great interest to Plos One readers and all that are interested in nwe methods of teching and learning.

Reviewer #3: The paper is interesting, and the information is up-to-date. However, the article has been conducted in a poor manner. It should be improved in the following directions:

[1] The paper must be fully developed - includes discussion, contribution, implication, and limitations.

[2] I would like to see a well-developed discussion (minimum two pages) comparing and contrasting solution/results presented in the work with existing work and then a subsection of it presenting contributions to theory/knowledge/literature (at least one to two paragraphs) and followed by a subsection on Implications for practice (at least one page). In these paragraphs authors should compare their research approach with previous research, citing references of others' research.

[3] The overall document should be checked for grammar, syntax and typos errors. Based on the above comments, I strongly believe that the authors will improve the quality of their manuscript given that they will make a detailed revision of the manuscript based on the provided comments

[4] The authors must add a new section called Background Literature which establishes the context of the research. This section explains why this particular research topic is important and essential to understanding the main aspects of the study. Usually, the background forms the first section of a research article/thesis and justifies the need for conducting the study and summarizes what the study aims to achieve.

[5] The authors must add a pseudocode diagram in section 2 explaining how the software algorithm is implemented. What software is used to implement the VR platform?

[6] In the Section 2, the authors should briefly highlight the main developments of their research topic and identify the main gaps that need to be addressed. In other words, this section should give an overview of your study. The section should be organized as: What is known about the broad topic?, What are the gaps or missing links that need to be addressed?, What is the significance of addressing those gaps?, and What are the rationale and hypothesis of your study?. Please rewrite this section.

6. PLOS authors have the option to publish the peer review history of their article (what does this mean?). If published, this will include your full peer review and any attached files.

Reviewer #1: **Yes: **Nikolaos Pellas

Reviewer #2: No

Reviewer #3: No

---

## [Author Response · Author response to Decision Letter 0]

12 May 2023

We thank all the reviewers for their careful consideration and insightful comments. We have updated our manuscript in accordance with these comments, as explained below. In the main text, updated portions are indicated in red.

Reviewer #1: 

1. Research questions, that drive the paper, should be built in the introduction from an ongoing and pertinent bibliography (up to 2022-23) and these should be of global interest and not focused on a particular local problem. Identifying a research gap is the most important by indicating in-text some newer references that are significant to your particular field of research.

2. The problem statement needs to be clear into Introduction. Does the relevant literature refer the potentials of using “conventional” instead of typical AR-supported activities to drive us there?

3. Some indicative literature review articles regarding the use of AR-supported learning that the same authors read maybe of great importance to readers in order to identify the research “gap” more easily without mention too many previous studies.

4. The authors need to provide any potentials regarding the use of AR settings. Are there any literature reviews to give us a point of view about this investigation? In other words, the authors need to “combine” any key word that investigate with any relevant study that can be integrated in-text as “background”.

In the Introduction section, we have added references to several of the latest studies regarding conventional affective communication intervention, such as [6], [11], and [12]. In addition, the problem statement (research question) paragraph has been added in the Introduction as well. Furthermore, we introduced the latest survey papers regarding AR-based skill training such as [13]-[16], which include several application scenarios using AR for training.

5. Any information respecting participants’ background can be beneficial for readers to understand better the level of technological literacy can overcome this “novelty effect” using AR.

We have added a description stating the participants had average computing literacy but no experience using VR/AR.

6. The authors should make explicit suggestions about how their study affects the design or use of AR systems. Is there something new about a particular theory, or is there evidence of theory advancement?

In the revised version of our paper, we have introduced the concept and detailed flowchart of how the agent responds in section 2. We believe this can serve as a design reference for the development of future systems.

7. The added value of authors’ work is not clear in terms of proper and current research (up to 2022-23), because several references are not up-to-date.

8. Authors should answer the main research question in the conclusions and discussion. Please provide a reasonable need to read your work’s results than previous ones or simply answer what we learned compared with current, significant research (up to 2022 should be your work’s “significance”).

In the updated version, we have introduced the latest related studies (including surveys) and described the position of our paper in the Introduction and Discussion sections.

9. How general are your results and how do you believe that such findings have to be of global interest? Please relate these with your limitations and Discussion that is not exist. Why?

10. Are there any points of view related to the consequences of this study’s limitations that may have an impact on their findings?

11. Conclusions and limits are too short for such a study.

12. Practical and educational implications are not provided.

We have added a section regarding the limitation of the number of participants and duration of the intervention. However, as we have written, our work has a relatively large number of participants in this sort of study using AR. Regarding the potential of our work, we have added the position of the paper in the Discussion section.

Reviewer #2:

This is a well written paper that presents a relevant description of a new methodology. The authors useful information has been provided. this papare may be of great interest to Plos One readers and all that are interested in nwe methods of teching and learning.

Thank you very much for kindly understanding the potential of the paper.

Reviewer #3: 

[2] I would like to see a well-developed discussion (minimum two pages) comparing and contrasting solution/results presented in the work with existing work and then a subsection of it presenting contributions to theory/knowledge/literature (at least one to two paragraphs) and followed by a subsection on Implications for practice (at least one page). In these paragraphs authors should compare their research approach with previous research, citing references of others' research.

[4] The authors must add a new section called Background Literature which establishes the context of the research. This section explains why this particular research topic is important and essential to understanding the main aspects of the study. Usually, the background forms the first section of a research article/thesis and justifies the need for conducting the study and summarizes what the study aims to achieve.

As other reviewers have provided similar comments, we have added the introduction, discussion, and limitation sections to meet your request, though we did not follow the exact format suggested. We introduced more recent related studies and described the position of our paper as well as the limitations. 

[3] The overall document should be checked for grammar, syntax and typos errors. Based on the above comments, I strongly believe that the authors will improve the quality of their manuscript given that they will make a detailed revision of the manuscript based on the provided comments

Thank you for the advice. We will carefully proofread and have the manuscript checked by external proofreaders.

[5] The authors must add a pseudocode diagram in section 2 explaining how the software algorithm is implemented. What software is used to implement the VR platform?

We have added the flowchart of the avatar’s motion in section 2. We used a Unity 2020 environment for the development.

[6] In the Section 2, the authors should briefly highlight the main developments of their research topic and identify the main gaps that need to be addressed. In other words, this section should give an overview of your study. The section should be organized as: What is known about the broad topic?, What are the gaps or missing links that need to be addressed?, What is the significance of addressing those gaps?, and What are the rationale and hypothesis of your study?. Please rewrite this section.

We have described our aim and hypotheses at the end of the first section (Introduction), specifically in lines 12-43, page 2. We also introduced existing literature regarding affective communication training and outcomes and VR/AR systems. Lastly, we described our research questions and hypotheses in lines 53-73.

---

## [Decision Letter · Decision Letter 1]

22 Jun 2023

Augmented reality-based affective training for improving care communication skill and empathy

PONE-D-22-35280R1

Dear Dr. Nakazawa,

We’re pleased to inform you that your manuscript has been judged scientifically suitable for publication and will be formally accepted for publication once it meets all outstanding technical requirements.

Kind regards,

Humaira Nisar

Academic Editor

PLOS ONE

Additional Editor Comments (optional):

Reviewers' comments:

Reviewer's Responses to Questions

**Comments to the Author**

1. If the authors have adequately addressed your comments raised in a previous round of review and you feel that this manuscript is now acceptable for publication, you may indicate that here to bypass the “Comments to the Author” section, enter your conflict of interest statement in the “Confidential to Editor” section, and submit your "Accept" recommendation.

Reviewer #1: All comments have been addressed

Reviewer #2: All comments have been addressed

Reviewer #3: All comments have been addressed

2. Is the manuscript technically sound, and do the data support the conclusions?

Reviewer #1: Yes

Reviewer #2: Yes

Reviewer #3: Yes

3. Has the statistical analysis been performed appropriately and rigorously? 

Reviewer #1: Yes

Reviewer #2: N/A

Reviewer #3: Yes

4. Have the authors made all data underlying the findings in their manuscript fully available?

Reviewer #1: Yes

Reviewer #2: Yes

Reviewer #3: Yes

5. Is the manuscript presented in an intelligible fashion and written in standard English?

Reviewer #1: Yes

Reviewer #2: Yes

Reviewer #3: Yes

6. Review Comments to the Author

Reviewer #1: The authors have satisfactorily responded to all questions and made the necessary changes to the manuscript.

Reviewer #2: (No Response)

Reviewer #3: I am pleased to note that the authors have adequately addressed all the concerns raised in my previous comments. They have taken the necessary steps to improve the paper by addressing the issues related to the introduction, methodology, and the use of programming code figures. The expanded introduction now provides a better context for the research, the methodology section has been elaborated to provide a clearer understanding of the procedures employed. These revisions greatly enhance the clarity and comprehensibility of the paper, making it a more valuable contribution to the field. I commend the authors for their efforts in addressing the concerns and improving the quality of their work.

7. PLOS authors have the option to publish the peer review history of their article (what does this mean?). If published, this will include your full peer review and any attached files.

Reviewer #1: No

Reviewer #2: No

Reviewer #3: No

---

## [Editor Report · Acceptance letter]

27 Jun 2023

PONE-D-22-35280R1 

Augmented reality-based affective training for improving care communication skill and empathy 

Dear Dr. Nakazawa:

I'm pleased to inform you that your manuscript has been deemed suitable for publication in PLOS ONE. Congratulations! Your manuscript is now with our production department. 

Kind regards, 

on behalf of

Dr. Humaira Nisar 

Academic Editor

PLOS ONE